# Identification of a Selective RelA Inhibitor Based on DSE-FRET Screening Methods

**DOI:** 10.3390/ijms21239150

**Published:** 2020-11-30

**Authors:** Yoshitomo Shiroma, Go Fujita, Takuya Yamamoto, Ryou-u Takahashi, Ashutosh Kumar, Kam Y. J. Zhang, Akihiro Ito, Hiroyuki Osada, Minoru Yoshida, Hidetoshi Tahara

**Affiliations:** 1Department of Cellular and Molecular Biology, Graduate School of Biomedical and Health Sciences, Hiroshima University, 1-2-3, Kasumi, Minami-ku, Hiroshima 734-8553, Japan; shiroma1023@hiroshima-u.ac.jp (Y.S.); Kaqd-80do@yahoo.co.jp (G.F.); Yamamoto.takuya@fukurou.ch (T.Y.); rytakaha@hiroshima-u.ac.jp (R.-uT.); 2Laboratory for Structural Bioinformatics, Center for Biosystems Dynamics Research, RIKEN, 1-7-22 Suehiro, Tsurumi, Yokohama, Kanagawa 230-0045, Japan; akumar@riken.jp (A.K.); kamzhang@riken.jp (K.Y.J.Z.); 3Laboratory of Cell Signaling, School of Life Sciences, Tokyo University of Pharmacy and Life Sciences, 1432-1, Horinouchi, Hachioji, Tokyo 192-0392, Japan.; aito@toyaku.ac.jp; 4Chemical Genomics Research Group, RIKEN Center for Sustainable Resource Science, 2-1 Hirosawa, Wako, Saitama 351-0198, Japan; yoshidam@riken.jp; 5Chemical Biology Research Group, RIKEN Center for Sustainable Resource Science, 2-1 Hirosawa, Wako, Saitama 351-0198, Japan; osadahiro@riken.jp; 6Seed Compounds Exploratory Unit for Drug Discovery Platform, RIKEN Center for Sustainable Resource Science, 2-1 Hirosawa, Wako, Saitama 351-0198, Japan; 7The Research Center for Drug Development and Biomarker Discovery, Hiroshima University, 1-2-3, Kasumi, Minami-ku, Hiroshima 734-8553, Japan

**Keywords:** NF-κB, DNA-binding protein, drug discovery, high-throughput screening

## Abstract

Nuclear factor-κB (NF-κB) is an important transcription factor involved in various biological functions, including tumorigenesis. Hence, NF-κB has attracted attention as a target factor for cancer treatment, leading to the development of several inhibitors. However, existing NF-κB inhibitors do not discriminate between its subunits, namely, RelA, RelB, cRel, p50, and p52. Conventional methods used to evaluate interactions between transcription factors and DNA, such as electrophoretic mobility shift assay and luciferase assays, are unsuitable for high-throughput screening (HTS) and cannot distinguish NF-κB subunits. We developed a HTS method named DNA strand exchange fluorescence resonance energy transfer (DSE-FRET). This assay is suitable for HTS and can discriminate a NF-κB subunit. Using DSE-FRET, we searched for RelA-specific inhibitors and verified RelA inhibition for 32,955 compounds. The compound A55 (2-(3-carbamoyl-6-hydroxy-4-methyl-2-oxopyridin-1(2H)-yl) acetic acid) selectively inhibited RelA–DNA binding. We propose that A55 is a seed compound for RelA-specific inhibition and could be used in clinical applications.

## 1. Introduction

Nuclear factor-κB (NF-κB) is a transcription factor that stimulates the expression of its target gene in response to stimuli, including cytokines IL-2 and TNF-α, viral or bacterial antigens, and UV irradiation. NF-κB plays an important role in biological functions controlling various physiological functions, such as innate and adaptive immunity, inflammation, stress response, and B cell maturation [1]. Interestingly, constitutive activation of NF-κB proteins has been observed in various types of tumors and is known to contribute to the malignant transformation of cancer. Reports have indicated that NF-κB can activate the transcription of genes associated with infiltration and metastasis, such as the matrix metalloproteinases (*MMPs*), urokinase types of plasminogen activator (*uPA*), and interleukin-8 (*IL-8)* [2,3,4]. Furthermore, NF-κB activates the cell cycle via the transcriptional regulation of cyclin D1 and c-MYC [5,6]. Further, NF-κB activates the transcription of the angiogenic factor vascular endothelial growth factor (VEGF) [7], and antiapoptotic genes such as the inhibitors of apoptosis proteins (*IAP*) [8,9] and fas-associated protein with death domain (FADD)-like IL-1β converting enzyme inhibitory protein (*FLIP*) [10]. NF-κB may be important factor in the regulation of cancer progression; hence, it has attracted significant attention as a potential target in cancer therapy.

Drugs that have already been used clinically with some inhibitory activity against the NF-κB pathway include nonsteroidal anti-inflammatory drugs, such as aspirin or curcumin, one of the components of turmeric. Although their specificity is low, these drugs have been reported to exhibit anticancer effects [11,12,13]. Other drugs developed as NF-κB inhibitors include dehydroxymethylepoxyquinomicin (DHMEQ) and NF-κB decoy oligos. DHMEQ, discovered by Umezawa et al., directly binds to the Rel homology domain (RHD) necessary for DNA binding and dimerization in the NF-κB family (e.g., p50, RelA, RelB, and cRel), thus inhibiting nuclear translocation and DNA binding [14]. This compound was demonstrated to have antitumor effects in mice that were transplanted with glioblastoma [15]. NF-κB decoy oligos can suppress the transcription activity of NF-κB by competitively inhibiting its DNA binding [16]. Therefore, existing NF-κB inhibitors are drugs that affect the entire NF-κB family, while protein-selective drugs have not yet been developed.

NF-κB has five subunits—RelA, RelB, cRel, p50, and p52—functioning primarily as heterodimers of p50–RelA or p52–RelB. The p50–RelA heterodimer-mediated transcriptional activation pathway is referred to as the canonical pathway and activates the transcription of genes involved in survival, proliferation, inflammation, and innate immunity [17]. In contrast, the noncanonical pathway through p52–RelB activates the transcription of genes involved in acquired immunity, such as B cell maturation and lymphogenesis [18]. For this reason, inhibition of the entire NF-κB pathway alters the regulation of various types of genes and may have unexpected consequences. The luciferase reporter assay, primarily used as a conventional screening method, is a superior technique that can be amenable to high-throughput screening (HTS), though it lacks the ability of discrimination against various proteins. For example, when RelA is inhibited, other NF-κB proteins bind to the recognition sequence and increase the expression of luciferase as NF-κB proteins have the same recognition sequence. Therefore, we developed a DNA strand exchange fluorescence resonance energy transfer (DSE-FRET) assay that can assess protein–DNA binding in real-time by combining DNA strand exchange reactions and the FRET reaction [19]. This method enables HTS using individual proteins and is suitable for finding a protein-specific inhibitor among a group of proteins with the same recognition sequence.

Herein, we demonstrate the use of DSE-FRET to search for RelA-specific inhibitors. Of the NF-κBs, RelA is highly expressed in pancreatic ductal adenocarcinoma (PDAC) patients and is associated with carcinogenesis and malignant transformation [20,21]. It has also been reported that gemcitabine-resistant cell lines lose their resistance to RelA due to siRNA treatment [22]. Thus, compounds with low molecular weight that inhibit RelA may be used as drugs to prevent treatment resistance in pancreatic cancer. We identified compound A55, a selective RelA inhibitor, using in silico and DSE-FRET assay screening. In the future, this compound could be used as a seed for selective RelA inhibitors.

## 2. Results

### 2.1. Screening for RelA-Specific Inhibitors Using in Silico Methods and DSE-FRET

We first conducted a two-sided HTS to identify RelA-selective inhibitors using a large number of compounds in silico (Figure 1). For HTS, we conducted DSE-FRET assays with a library of 32,914 compounds (Tokyo University Core Library, RIKEN Natural Products Depository (RIKEN NPDepo), known drugs library) at a concentration of 10 µM or 5 µg/mL, with dimethyl sulfoxide (DMSO) serving as a negative control. We calculated the Z’ [19] factor for each plate to investigate whether an assessment was correctly conducted, and a Z’ factor of 0.7 or higher was obtained for the plates (Appendix A). Suramin and NP322 were identified from this screen, both suppressing RelA–DNA binding by over 30%, p52–DNA binding by less than 10%, and autofluorescence by less than 2.5 (Figure 1a, Appendix A).

For in silico screening, we constructed a model of RelA–DNA interaction with small molecules. Small molecular modulation of the RelA–DNA interface using in silico screening methods is challenging because of the lack of information on druggable pockets, corroborated by previous reports encountering difficulty in targeting transcription factors due to their shallow binding pockets [23,24]. Hence, to identify suitable druggable pockets on the surface of RelA, putative small-molecule binding pockets were predicted. Due to slight structural differences between the two monomers (RMSD 2.8 Å), the RelA homodimer was used for pocket prediction. Four out of five predicted pockets revealed a SiteScore equal to or greater than 0.8 (Table 1), which is a generally accepted cutoff for druggable pockets [25,26]. The top scoring pocket—with a SiteScore value of 0.97 (red spheres, Appendix A)—included loop L1 amino acid residues adjacent to the DNA-binding interface. In the RelA crystal structure bound to DNA (PDB code 2RAM), these residues have been shown to occupy the major groove of the DNA [27]. Interestingly, Cys38 and Cys120 near this pocket have been previously targeted for the development of a covalent inhibitor to the RelA–DNA interaction [28,29,30,31]. The next two pockets (Pockets 2 and 3) refer to the same pocket, each on one of the RelA monomers. This pocket consists of residues that are in direct contact with DNA (Table 1, Appendix A). The other two pockets were located further away from the DNA-binding interface, thus seemingly unsuitable for virtual screening.

Two separate virtual screenings were conducted, targeting the two predicted binding pockets—one pocket from each RelA monomer, pockets 1 and 3, respectively—following a hierarchical multistep docking protocol (Figure 1b). One hundred hits from two separate docking runs were merged, the duplicates were removed, and visual analysis of their interaction with the binding pocket was performed. Interactions with positively charged residues at the DNA-binding interface, such as Arg35 or Arg41 from chain B, were prioritized. Finally, a set of 50 compounds was selected for purchase, of which 41 were obtained (Figure 1b).

DSE-FRET was conducted in a 96-well plate to investigate the RelA–DNA binding inhibition of these 41 compounds. The results showed that A55 (2-(3-carbamoyl-6-hydroxy-4-methyl-2-oxopyridin-1(2H)-yl) acetic acid) had an autofluorescence of less than 2.5 and a high RelA–DNA binding effect (over 50%). The other 40 compounds had either high autofluorescence or weak inhibitory effects (Appendix A).

Validation was conducted at a compound concentration of 50 µM to investigate whether high compound concentrations have an effect on p52–DNA binding. DSE-FRET was conducted in 96-well plates using suramin, NP322, and A55. Suramin and NP322 inhibited both RelA–DNA and p52–DNA binding, whereas A55 inhibited only RelA–DNA binding (Figure 1c, Appendix A). In addition, we performed electrophoretic mobility shift assays (EMSA) using the recombinant proteins of RelA or p52 and showed that A55 selectively inhibited the RelA–DNA binding (Appendix A). Therefore, A55 was identified as a candidate RelA-selective inhibitor. 

### 2.2. A55 Binds to and Inhibits RelA 

DSE-FRET was conducted using compound concentrations of 2–10 µM to investigate whether the effects of A55 were concentration-dependent, or whether this compound exhibited inhibitory effects at low concentrations. The results showed that A55 inhibited RelA–DNA binding in a concentration-dependent manner (Figure 2a). The binding dissociation constants of glutathione S-transferase (GST)–RelA, GST–p52, and A55 were investigated using surface plasmon resonance (SPR) analysis. The Rmax and dissociation constant (Kd) for 0.625–20 µM A55 were 7.3 and 2.87 µM, respectively, and 4.7 and 13.0 µM, respectively, for 3.125–100 µM A55 in GST–RelA (Figure 2b, Appendix A). The theoretical and actual Rmax values were close to one another, indicating that RelA and A55 have a one-to-one relationship. Further, in GST–p52, the Rmax and Kd values for 0.625–20 µM A55 were not determined and were found to be 11.4 and 1.27 µM for 3.125–100 µM A55, respectively (Figure 2c, Appendix A). Overall, A55 appears to closely bind RelA but weakly bind p52.

### 2.3. A55 Is a RelA-Specific Inhibitor

To investigate the inhibitory effects of NF-κB proteins other than RelA, we calculated the half maximal inhibitory concentration (IC50) for the inhibition of RelA, cRel, p50, and p52 by A55. For the NF-κB subunits, DSE-FRET assessment was not possible for the homodimer of RelB as it does not have DNA-binding capabilities [32]. The IC50 of RelA was 9.31 µM, whereas those of other proteins exceeded 100 µM (Figure 3a). The inhibition rate at 10 µM is shown in Figure 3b. We also conducted EMSA using nuclear extract from the pancreatic cancer cell line PK8 to determine whether the inhibitory effect of RelA would be observed in the presence of other proteins. NF-κB is activated by TNF-α stimulation and translocated into the nucleus, where it binds to DNA [33]. Therefore, PK8 was first treated with TNF-α (50 ng/mL) to activate NF-κB, followed by extraction of the nucleus 10 min later (Appendix A). EMSA revealed two band shifts (lane 2), though both bands disappeared when a competitor containing an NF-κB sequence was included (lane 3), indicating that NF-κB was indeed bound. Furthermore, the upper band (lane 6) was supershifted due to RelA antibodies, confirming that the upper band was RelA. Only the upper band disappeared in the presence of A55 (lanes 7–10), indicating that DNA by RelA was inhibited (Figure 3c). Together, these results demonstrate that the compound A55 is a selective inhibitor of RelA.

### 2.4. Docking Predictions 

We conducted in silico prediction to determine how A55 binds with RelA. The docking-predicted binding mode of A55 suggests that the compound binds to a region near loop L1 residues through a number of hydrogen bond interactions (Figure 3). Specifically, the carbonyl group at the 2-position in the pyridine ring forms two hydrogen bonds with the Arg41 sidechain. Further, the backbone carbonyl of Arg41 is also involved in hydrogen bond contact with the carbamoyl nitrogen. Arg41 is a key loop L1 residue and plays an important role in DNA interactions [27]. Additionally, hydrogen bonds between the hydroxy and acetate substituents at the pyridine ring and Gln119 sidechain appear to further stabilize A55.

## 3. Discussion

In the present study, we identified A55 as a RelA inhibitor using DSE-FRET assay with 32,955 compounds (Figure 1). In addition, DSE-FRET assay suggested that A55 is a selective RelA inhibitor, as shown in Figure 3a, although we could not analyze inhibition of A55 to RelB–DNA interaction as the homodimer of RelB does not have DNA-binding capabilities [32]. The interpretation was supported by SPR and EMSA analysis. SPR analysis showed that A55 closely binds RelA but weakly binds p52 (Figure 2b,c). EMSA analysis demonstrated that A55 interferes with only RelA–DNA complex (upper band) and not other protein–DNA complexes (lower band) (Figure 3c). Furthermore, 500 µM of A55 did not affect the lower band. These results suggest that A55 is a selective RelA inhibitor. 

This study also suggests that DSE-FRET can be used to search for inhibitors of various DNA-binding proteins. In fact, DSE-FRET on 32,914 compounds identified 21 compounds that suppressed RelA–DNA bonds (Appendix A), which could be utilized as NF-κB inhibitors in the future. In particular, suramin has been reported to have various inhibitory effects [34]. These results suggest that DSE-FRET may be applied to screen for drug repositioning.

RelA-specific inhibitors may contribute not only to research tools for NF-κB but also for cancer treatment. RelA is highly expressed in PDAC patients and has been shown to be involved in the malignant transformation of cancer [20,21]. It has been reported that the use of siRNA against RelA in gemcitabine-resistant cell lines decreases resistance [22]. These results show that RelA contributes to cancer chemotherapy resistance, and the selective RelA inhibitor A55 may be used as a concomitant therapy. However, RelA systematic knockout mice has been found to undergo apoptosis in the liver during the embryonic period, resulting in embryonic lethality, and RelA has also been associated with apoptosis in other tissues such as the nerves, skin, and intestinal tract [35,36]. Thus, we need to evaluate the side effects of the RelA inhibitor. As the present study focused on identification of the inhibitor, animal studies remain warranted.

To evaluate the cytotoxicity and anticancer effect of A55, we examined the cell proliferation of TNF-α-stimulated PK8 cells in the presence or absence of A55. Cell proliferation assay revealed that A55 enhanced the growth inhibition of TNF-α-treated PK8 cells in a dose-dependent manner (Appendix A). A55 has several hydrophilic groups and a log*P* value of −1.78, which is the log value of the octanol/water partition coefficient, *P*, due to its low hydrophobicity and cell membrane permeability. Therefore, high-concentration of A55 is required for cell growth inhibition by TNF-α treatment; further improving the cell permeability or hydrophobicity of A55 is essential. Like DHMEQ derived from epoxyquinomicin C [37], it may be possible to create a stronger RelA-specific inhibitor from A55. While A55 has moderate potency due to its low molecular weight and only 16 heavy atoms, its ligand efficiency is significantly high (LE = 0.44). Moreover, a similarity search against the ChEMBL [38] database using A55 revealed no known biological activity for this compound. Evaluation of this compound for pan-assay interference compounds (PAINS) [39] and problematic compounds according to the Eli Lilly MedChem rules [40] revealed no structural alternation. This suggests the significant potential of this compound.

It was predicted during in silico analysis that hydrogen bonds would be formed in the Arg41 portion of RelA and immobilized in the Gln119 section (Figure 4). Although docking calculations were unable to pinpoint the exact mechanism by which A55 interferes with RelA–DNA interactions, we hypothesize that compound binding might disrupt the key contacts with DNA by either altering the L1 loop conformation or by making contacts unfavorable for DNA binding. Detailed experiments using X-ray crystal structure analysis are warranted for more accurate information.

To our knowledge, this is the first report on selective RelA inhibitors. The development of this compound will advance NF-κB research and may contribute to cancer therapy.

## 4. Materials and Methods 

### 4.1. Structure-Based Virtual Screening to Identify RelA–DNA Interaction Inhibitors

To identify compounds interfering with the interaction between RelA and DNA, structure-based virtual screening was carried out following a hierarchical docking protocol [41]. Putative small-molecule binding pockets on the surface of RelA were predicted using SiteMap [25,26]. The crystal structure of the RelA homodimer in complex with the DNA (PDB code 2RAM) was used for predictions [27], with the DNA and water molecules removed and the structure prepared using Maestro (Schrodinger, Inc., New York, NY, USA). A standard grid of 1.0 Å was used for calculations. The top five binding pockets with at least 15 site points were saved.

The Namiki Shoji collection (http://www.namiki-s.co.jp) of approximately 4 million commercially available compounds was subjected to a hierarchical docking procedure to select a number of suitable candidates for biological evaluation. For structure-based virtual screening, the compounds in a small-molecule library were first docked to the putative small-molecule binding pockets on the surface of RelA using FRED [42,43]. The receptor for molecular docking was prepared using the “make_receptor” utility (OEDocking suite, OpenEye Scientific Software, Santa Fe, NM, USA). An ensemble of a maximum of 200 conformations per compound in the screening library was generated using OMEGA [44,45]. Docking was carried out using the “high” resolution mode, and a single pose per compound was saved. All compounds were rank-ordered using the Chemgauss4 scoring function. The 10,000 top ranking molecules were then redocked using Glide for exhaustive sampling [46,47,48]. Ligands for Glide docking were prepared using LigPrep and OPLS-2005 forcefield [49]. Schrodinger’s Maestro protein preparation utility was employed for receptor preparation. Grids for molecular docking were generated using SiteMap pockets, and molecular docking was performed using the standard (SP) and extra precision (XP) modes in Glide. Compounds were rank-ordered based on the Glide “docking score”. The 100 top ranking compounds were selected. All compounds were visually analyzed for the interactions they made, and 50 compounds were selected for purchase from commercial vendors.

### 4.2. Protein Preparation

The expression and purification of the GST-fusion recombinant human NF-κB proteins (RelA, RelB, cRel, p50, and p52) were conducted as previously described [30]. The necessary vectors were gifted by Dr. Umezawa (Aichi Medical University).

### 4.3. DSE-FRET

The technique for DSE-FRET and duplexes have been described previously [19]. Oligonucleotides 5′-FAG TTG AGG GGA CTT TCC CAG GCG ACT CAC TAT AGG Cgg tgt ctc gct cgc-3′ (NF-01F), 5′-AGT TGA GGG GAC TTT CCC AGG CGA CTC ACT ATA GGc acc aca cca ttc cc-3′ (NF-13), 5′-ggg aat ggt gtg CCT ATA GTG AGT CGC CTG GGA AAG TCC CCT CAA CTD-3′ (NF-14D), and 5′-gcg agc gag aca ccG CCT ATA GTG AGT CGC CTG GGA AAG TCC CCT CAA CT-3′ (NF-02) were used for NF-κB-DSE-FRET (F, 6FAM; D, DABCYL; lower-case letters, single-stranded tail; underline, NF-κB binding site). Plate-based (384 scale) DNA-binding experiments were carried out with 0.1 µL of 2 mM or 1 mg/mL compound, 2 µL of 80 nM NF-D1 (consisting of NF-01F and NF-14D), 1 µL of 20 ng/µL poly dI-dC, and 6.9 µL of 3.2 pmol protein in binding buffer (10 mM 4-(2-hydroxyethyl)-1-piperazineethanesulfonic acid (HEPES)–NaOH pH 7.9, 50 mM KCl, 0.1 mM ethylenediaminetetraacetic acid (EDTA), 2.5 mM dithiothreitol (DTT), 10% glycerol, and 0.05% Nonidet P40). The plates were incubated for 30 min at room temperature, after which 10 µL of NF-D2 (consisting of NF-02 and NF-13) was added. The plates were centrifuged at 1000 rpm for 30 s, followed by shaking for 30 s using a plate shaker (Biotec). Fluorescence was measured at 528 nm, after excitation at 485 nm, using a multiplate reader (Biotec) over a period of 60 min. 

To obtain the inhibition rate, the following equation was solved using the “solver” add-on (Microsoft Excel):
$$\%\;inhibition=\frac{\left(\Delta A-\Delta B\right)}{\left(\Delta C-\Delta B\right)}\times 100$$
where *A* is the observed fluorescence at sample treatment, *B* is the observed fluorescence at solvent treatment, and *C* is the observed fluorescence of free protein. Δ indicates that the last fluorescence intensity was subtracted from the first fluorescence intensity.

Autofluorescence was obtained as follows:
$$Autofluorescence\;=\frac{A}{B}$$
where *A* is the fluorescence value of the mixture with the compound and NF-D1, and *B* is the fluorescence value of NF-D1 without the compound.

### 4.4. SPR Analysis

SPR biosensor measurements were performed at 25 °C using a Biacore T100 instrument (GE Healthcare, Chicago, IL, USA). Anti-GST antibody (GE Healthcare, BR-1002-23) was immobilized on the CM5 sensor chip (GE Healthcare) using amine coupling. Next, 4 µg/mL GST–RelA or 5 µg/mL GST (GE Healthcare) was flowed through the channel (5 μL/min, for 180 s) and captured on the sensor chip. The flow path was washed with 50 mM NaOH and equilibrated with running buffer (10 mM HEPES (pH 8.0), 60 mM KCl, 100 µM EDTA, 1% DMSO, and 0.0125% NP40). A55 (0.625–100 µM) was flowed through, and its interaction with GST–RelA or GST was measured. Dissociation constants were computed by fitting to a 1:1 interaction model using Biacore (GE Healthcare). Stoichiometry was calculated based on the theoretical Rmax value, where the theoretical Rmax = MWA/MWL RL SM (A, analyte; L, ligand; RL, immobilization level of ligand RU; SM, stoichiometry). 

### 4.5. Cell Culture Work and Nuclear Protein Isolation

The human pancreatic carcinoma cell line PK8 was grown in Roswell Park Memorial Institute 1640 Medium (RPMI) (SIGMA; R8758) supplemented with 10% fetal bovine serum (FBS) at 37 °C with 5% CO_2_. 

For EMSA using nuclear extract, PK8 cells were treated with TNF-α (50 ng/mL) for 10 min before nuclear protein extraction. The cells were washed with ice-cold phosphate-buffered saline (PBS), followed by buffer A (10 mM HEPES–NaOH, pH 7.9, 10 mM KCl, 1.5 mM MgCl_2_, 0.5 mM DTT, and 0.5 mM phenylmethanesulfonyl fluoride (PMSF)). After 15 min on ice, the cell suspension was centrifuged at 4 °C for 3 min at 2000 rpm, and the resultant cell pellets were resuspended in buffer A. Cells were homogenized using Dounse homogenizer type A (tight, 50 stroke) on ice. Homogenates were centrifuged at 4 °C for 10 min at 5000 rpm, and the resultant nuclear protein pellets were resuspended in buffer C (20 mM HEPES–NaOH pH 7.9, 600 mM KCl, 1.5 mM MgCl_2_, 25% glycerol, 0.2 mM EDTA, 0.5 mM DTT, and 0.5 mM PMSF). The nuclear proteins were incubated for 1 h on ice with intermittent tapping followed by 15 min of centrifugation at 18,000 rpm (4 °C). The supernatants were dialyzed using a Slide-A-Layzer Dialysis cassette 3500MW (Thermo: 66110) with buffer D (20 mM HEPES–NaOH pH 7.9, 100 mM KCl, 20% glycerol, 0.2 mM EDTA, 0.5 mM DTT, and 0.5 mM PMSF). The dialysis products were then aliquoted, snap-frozen, and stored at −80 °C. Quantitation of nuclear protein content was performed using an XL-Bradford assay (APRO: KY-1031).

### 4.6. SDS PAGE and Western Blot Analysis 

The nuclear extracts were boiled for 5 min in sodium dodecyl sulfate (SDS) buffer (0.12 M Tris-HCl pH 6.8, 10% glycerol, 3.7% SDS, 0.2 M DTT, and 0.004% bromophenol blue), fractionated by SDS gel electrophoresis, and transferred to Immobilon-P membranes (Millipore, Burlington, MA, USA: IPVH00010). Membranes were blocked for 5 min at room temperature with 5% nonfat dry milk in Tris-buffered saline (TBS: 10 mM Tris base, 40 mM Tris-HCl, and 150 mM NaCl) containing 0.1% Tween-20 (TBST). Blots were incubated for 3 h at room temperature or overnight at 4 °C with antibodies against Ku70 (Santa Cruz, Dallas, Texas, USA: sc1486) 1:1000, Cdk4 (Santa Cruz: sc260) 1:1000, RelA (CST: 8242) 1:2000, RelB (GeneTex, Irvine, CA, USA: GTX102333) 1:500, cRel (GeneTex: GTX113264) 1:500, p50 (GeneTex: GTX100772) 1:500, or p52 (Millipore: #05-361) 1:1000, followed by incubation for 1 h with horseradish peroxidase (HRP)-linked rabbit or mouse antibody (Jackson Immuno Research, West Grove, PA, USA: 111-035-003/115-035-003). Immunoreactive bands were visualized by enhanced chemiluminescence reagent treatment and exposure to ECL western blotting detection reagent (Perkin Elmer, Waltham, MA, USA).

### 4.7. EMSA

EMSA reactions were conducted according to the gel shift assay protocol (Promega). Briefly, we used γ32P-labeled (NF-κB) and nonlabeled (competitor 1) DNA duplexes, 5′-AGTTGAGGGGACTTTCCCAGGC-3′ (eurofin) (NF-κB consensus sequence is underlined), and nonlabeled oligo duplex SP1(competitor 2), 5′-ATTCGATCGGGGCGGGGCGAGC-3′ (eurofin). We used 0.1 pmol labeled oligonucleotide and PK8 nuclear protein (20 ng) or 5 pmol recombinant proteins RelA or p52 during the reaction. For the competition reaction, 25-fold of the unlabeled oligonucleotide was added 15 min prior to addition of the respective radiolabeled probe. For the supershift reaction, 1 µL of either rabbit polyclonal RelA antibody (Santacruz; sc-372) or rabbit IgG (sc-2027) was added 15 min before adding the respective radiolabeled probe. After incubation, the reaction mixtures were separated for 2 h on a 4% nondenaturing polyacrylamide gel at 50 V on AE-6450 (ATTO, Tokyo, Japan) using 0.5× tris-borate-EDTA (TBE). After electrophoresis, the gel was transferred to Whatman 3MM paper and dried in a vacuum gel dryer for 90 min at 80 °C. The dried gel was then exposed on an imaging plate and scanned using the FLA 7000 imaging system (Fujifilm, Tokyo, Japan).

### 4.8. Cell Viability Assay

Cell viability was measured using the PrestoBlue assay according to the standard protocol (Thermo Fisher Scientific, Waltham, MA, USA: A13261). PK8 cells were cultivated on 96-well plates. After a 48 h treatment with TNF-α (50 ng/mL) in the presence or absence of A55, the culture medium was replaced with fresh medium containing 10% PrestoBlue and incubated for an additional 1 h at 37 °C and 5% CO_2_. Fluorescence intensity was then measured using a multiplate reader (EnSpire, PerkinElmer, Waltham, MA, USA).

## Figures and Tables

**Figure 1 ijms-21-09150-f001:**
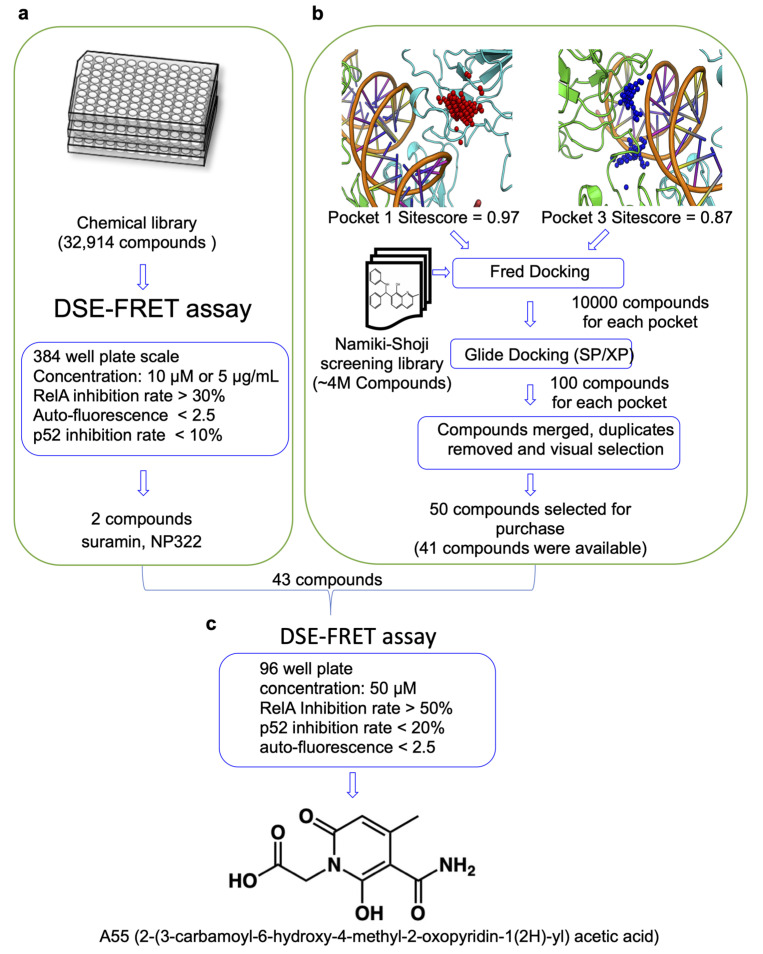
Screening for RelA-specific inhibitors using in silico and DNA strand exchange fluorescence resonance energy transfer (DSE-FRET) methods. (**a**) High-throughput screening (HTS) using DSE-FRET to identify selective RelA inhibitors. The inhibitory effects of RelA or p52 and the autofluorescence of 32,914 compounds were evaluated by DSE-FRET at 384-well plate scale. Two compounds were finally selected as hits. (**b**) Hierarchical structure-based virtual screening to identify compounds interfering with the RelA–DNA interaction. Hierarchical molecular docking was performed separately on the predicted small-molecule binding pockets (1 and 3) on the surface of each RelA monomer. Hits from both virtual screening were merged and ~50 molecules were selected based on diversity and the interactions they make with the binding pocket. (**c**) Retest and evaluation of RelA inhibitors. The inhibitory effects of RelA or p52 and the autofluorescence of suramin, NP322, and 41 available compounds at selected in silico screening were evaluated by DSE-FRET at 96-well plate scale.

**Figure 2 ijms-21-09150-f002:**
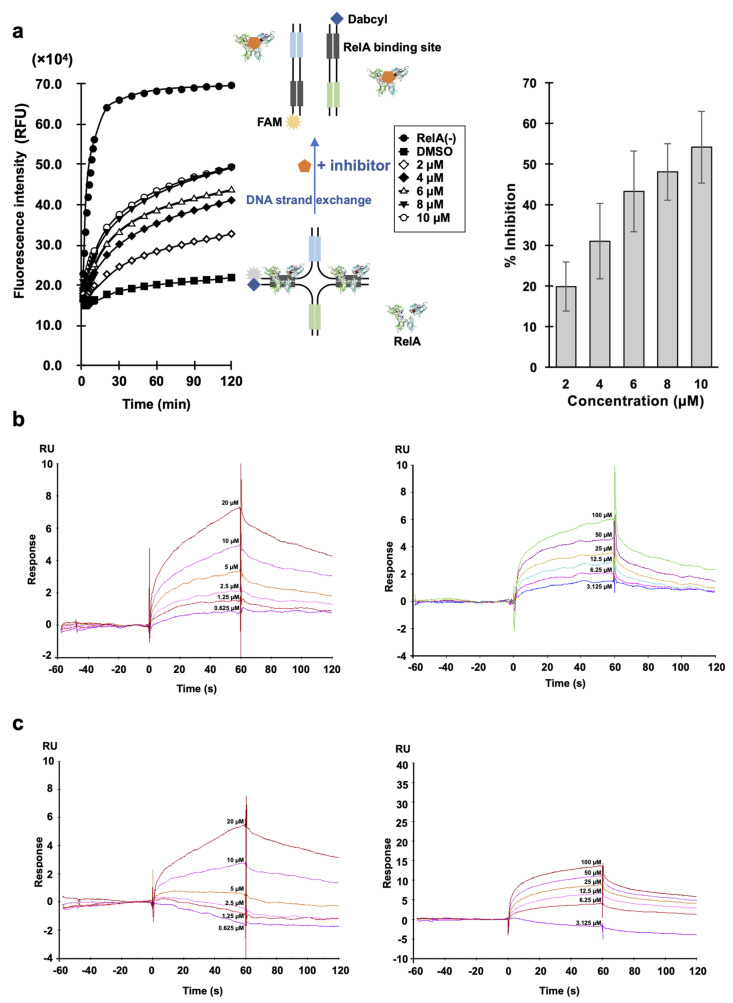
A55 binds to and inhibits RelA. (**a**) Dose–response of A55. Error bars represent SD (*n* = 3). (**b**,**c**) Interaction of A55 with immobilized glutathione S-transferase (GST)–RelA or GST–p52. The figure represents surface plasmon resonance (SPR) sensorgrams using 0.625–20 µM (left) and 3.125–100 µM (right) of A55 with immobilized GST–RelA (**b**) or GST–p52 (**c**).

**Figure 3 ijms-21-09150-f003:**
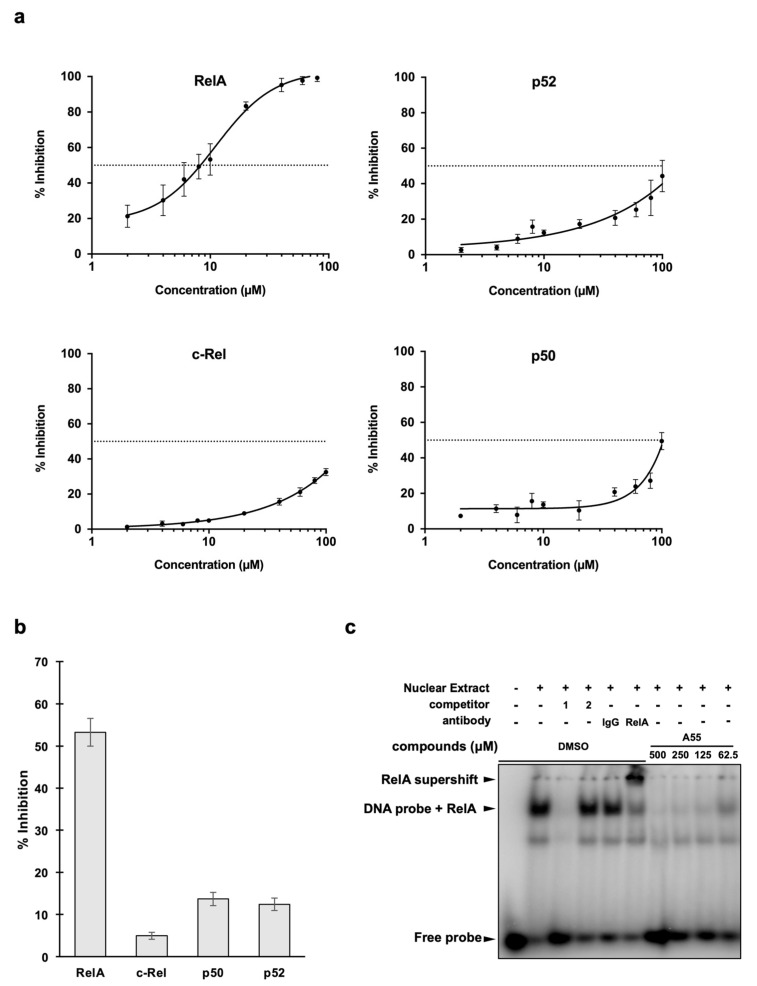
A55 is a RelA-specific inhibitor. (**a**) DSE-FRET inhibition rates of different concentrations of A55 against NF-κB RelA, p52, cRel, and p50. (**b**) DSE-FRET inhibition rates of 10 µM A55 against NF-κB proteins. Error bars represent SD (*n* = 3). (**c**) Electrophoretic mobility shift assay (EMSA) experiments with nuclear extracts from tumor necrosis factor (TNF)-α-treated PK8 cells were conducted using radiolabeled probes containing the NF-κB binding site. Free probe only (lane 1), PK8 nuclear extracts (lane 2), competitor experiments (competitor 1 is nonlabeled NF-κB probe (lane 3) and competitor 2 is nonlabeled sp1 probe (lane 4)), supershift experiments (rabbit IgG (lane 5) and rabbit RelA antibody (lane 6)), and different concentration of A55 (lanes 7–10).

**Figure 4 ijms-21-09150-f004:**
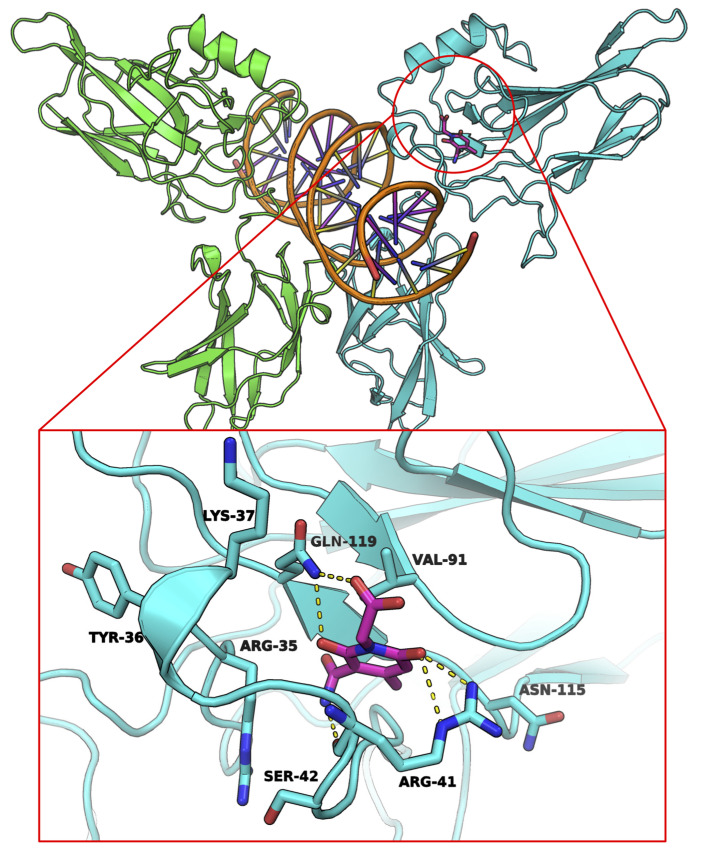
Molecular docking-predicted binding mode. Molecular docking-predicted binding mode of A55 (magenta atom colored sticks) to the RelA binding pocket near the DNA-binding interface. Yellow dashes represent hydrogen bonds between A55 and the amino acid residues of pocket 1.

**Table 1 ijms-21-09150-t001:** SiteMap predicted binding pockets on nuclear factor-κB (NF-κB) RelA surface.

Pocket #	Pocket Volume (Å^3^)	SiteScore	Chain	Residues
Pocket 1	137.8	0.97	B	Met32, Arg33, Phe34, Arg35, Tyr36, Lys37, Gly40, Arg41, Ala43, Gly44, Gly92, Lys93, Cys95, Asn115, Gly117, Ile118, Gln119
Pocket 2	212.3	0.92	B	Phe34, Tyr36, His88, Cys120, Val121, Lys122, Lys123, Leu126, Asp151, Tyr152, Leu154, Asn155, Asp185, Arg187, Ala188, Pro189, Asn190, Thr191, Ala192, Glu193, Leu194, Lys195, Asp217, Lys218, Val219, Gln220, Asp223, Ile224, Arg274
Pocket 3	194.1	0.87	A	Gly31, Arg33, Phe34, Tyr36, His88, Cys120, Val121, Lys122, Lys123, Leu154, Asn155, Asp185, Arg187, Ala188, Pro189, Asn190, Ala192, Glu193, Leu194, Lys195, Asp217, Lys218, Val219
Pocket 4	155.7	0.8	B	Thr71, Arg73, Ser75, Glu101, Arg133, Ile134, Asn137, Asn138, Asn139, Pro140, His142, Val143, Ile145, Gln162, Thr164, Leu174
Pocket 5	141.3	0.73	A	Arg30, Thr78, Lys79, Asp80, Pro82, Arg84, His86, Asp151, Tyr152, Asp153, Asn155, Ala156, Phe184, Asn190, Thr191, Arg274, Ser276, Asp277, Glu279

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
