# Peer review of "Identification of a Selective RelA Inhibitor Based on DSE-FRET Screening Methods"

_ijms, 2020, doi:10.3390/ijms21239150_

Round 1

Reviewer 1 Report

The authors screened 32,955 compounds that specifically inhibit RelA using the recently developed high-throughput screening (HTS) method, and identified the compound A55 (2-(3-carbamoyl-6-hydroxy-4-methyl-2-oxopyridin-1(2H)-yl) acetic acid). A55 selectively inhibited RelA DNA binding. The authors propose that A55 is a seed compound for RelA-specific inhibition and has potential for clinical application. The results of this study are very interesting and well analyzed. However, even if the purpose of this study is to identify selective inhibitors of RelA, some additional experiments are needed to confirm selectivity and safety.

  1. Is the font used in the text specified? The reviewer recognized that at least κ uses two different fonts. Check the font you used again.
  2. The authors showed the specificity of A55 with other NF-κB subunits, but make sure it does not suppress, for example, phosphorylation and degradation of IκBa or MAPK signaling.
  3. As the authors mentioned in the “Discussion” section, knock-in mice that block RelA DNA binding cause liver apoptosis similar to RelA knockout mice (Dong J et al. Genes Dev. 2008 22; 1159). Animal experiments are fine for future work, but at least examine cell death by stimulating PK8 cells with TNFa in the presence or absence of A55.
  4. References; The notation of κ is not unified, so please unify it.

Reviewer 2 Report

The current manuscript by Shiroma et al is focused on identifying and characterizing a selective RelA inhibitor based on a novel DSE-FRET high throughput screening (HTS) method. In the last two decades, several NF-kB inhibitors have been developed and been used in various clinical trials for the treatment of variety of inflammatory diseases and to increase the efficiency of conventional cancer therapy. However, the major problem with many of these inhibitors is the side effects because of its lack of selectivity. In the current manuscript present convincing data to show that A55 is selective to RelA-DNA binding (with IC50 of 9.31 µM). Data presented in this manuscript clearly suggest that A55 could be seed compound with selective RelA inhibition and has potential clinical application in future.

This reviewer has only one question. In the functional characterization experiment, authors have shown A55 interferes selectively binds with Rel A in EMSA after TNF treatment of PK8 cells. TNF is a potent activator of RelA-dependent canonical NF-kB pathway but not the non-canonical arm of the pathway. It would further strengthen their claim regarding selectivity of A55 if they can show lack of A55 binding to the activated p52/RelB DNA binding after non-canonical pathway activators such as BAFF or CD40.

Round 2

Reviewer 1 Report

The revised manuscript is well performed.